# Ionic liquid enables highly efficient low temperature desalination by directional solvent extraction

Jiaji Guo[1], Zachary D. Tucker[2], Yu Wang[1], Brandon L. Ashfeld ✉[2] & Tengfei Luo ✉[1,3]

Seawater desalination plays a critical role in addressing the global water shortage challenge. Directional Solvent Extraction (DSE) is an emerging non-membrane desalination technology that features the ability to utilize very low temperature waste heat (as low as 40 °C). This is enabled by the subtly balanced solubility properties of directional solvents, which do not dissolve in water but can dissolve water and reject salt ions. However, the low water yield of the state-of-the-art directional solvent (decanoic acid) significantly limits its throughput and energy efficiency. In this paper, we demonstrate that by using ionic liquid as a new directional solvent, saline water can be desalinated with much higher production rate and thus significantly lower the energy and exergy consumptions. The ionic liquid identified suitable for DSE is [emim][Tf$_2$N], which has a much (~10×) higher water yield than the currently used decanoic acid. Using molecular dynamics simulations with Gibbs free energy calculations, we reveal that water dissolving in [emim][Tf$_2$N] is energetically favorable, but it takes significant energy for [emim][Tf$_2$N] ions to dissolve in water. Our findings may significantly advance the DSE technology as a solution to the challenges in the global water-energy nexus.

[1] Department of Aerospace and Mechanical Engineering, University of Notre Dame, Notre Dame, IN 46556, USA. [2] Department of Chemistry and Biochemistry, University of Notre Dame, Notre Dame, IN 46556, USA. [3] Department of Chemical and Biomolecular Engineering, University of Notre Dame, Notre Dame, IN 46556, USA. ✉email: bashfeld@nd.edu; tluo@nd.edu

The shortage of viable water resources is rapidly reaching critical status on a global scale. While extended droughts in many areas is a contributing factor[1], industrial and residential pollution of regional and local water supplies exacerbates this growing crisis. Given that ocean and subterranean saline aquifers contain 97.5% of the global water, desalination is a promising means for meeting fresh water demand. While membrane-based desalination processes like reverse osmosis (RO) have drawn considerable attention, the need for high grid electricity renders their application in low-resource settings challenging[2]. RO can be energy-efficient (as low as 2 kWh/m$^3$) in centralized plants largely due to the implementation of mechanical energy recovery systems[3], but at smaller scales the energy cost can be much higher (up to 17 kWh/m$^3$)[4]. In contrast, directional solvent extraction (DSE), named one of the top ten world-changing ideas by Scientific American in 2012[5], is an attractive alternative as it requires comparatively low operation temperatures relying almost exclusively on the consumption of waste heat or unconcentrated solar energy[6,7]. The conceptual basis behind DSE centers on the use of a task-specific directional solvent (DS) that can solvate water in high yield, defined as the water solubility change per degree of temperature change (%/°C), is insoluble in water, and will not solvate salt ions[6–8]. It has been proposed that these directional solubilities are a result of the subtle balance between the hydrophilic and hydrophobic features of the solvent and the resulting intermolecular interactions with the solute[7]. Not surprisingly, a major impediment to the implementation of DSE processes is the identification of an optimal DS. Currently, decanoic acid, with a water yield of 0.027%/°C, represents the best performing DS to date[9]. This exceptionally low yield results in a low water production rate and relatively high energy consumption[6]. In contrast to aliphatic acids, we speculated that a readily functionalized heteroaromatic core would enable the desirable balance of hydrophobic and hydrophilic properties amenable to optimal DS performance. While tuning hydrophilicity of solids has been investigated that can contribute to thermal desalination[10–12], studies of liquid phase hydrophilicity for similar purposes have not been seen. Herein, we report the identification of an N-heterocyclic task-specific ionic liquid (TSIL) for DSE, discovered through the combined use of experiments and molecular dynamics (MD) simulations, that exhibits a ten-fold improvement in fresh water yield over decanoic acid.

Ionic liquids (ILs) have demonstrated an exceptional promise as a molecular framework for the development of task-specific fluids due to their design flexibility and inherently low volatility. ILs comprised of organic ions residing in a liquid state between room temperature and 100 °C have shown promise as working fluids across a variety of applications, including the separation of organic compounds, sequestration of transition metals, capturing carbon dioxide, and desalinating aqueous media[13–17]. The structural versatility and variability of many ILs enable extraordinary freedom in solvent design, as the cation and anion components can be individually engineered to achieve the desired directional solubilities. As illustrated in Fig. 1, an optimal DS IL would display sparing solubility with highly concentrated salt water (a), which then upon heating, draws only water into the IL while leaving the salts in the water phase yielding a high salinity brine (MX) (b). Removal of the MX byproduct (c) followed by a reduction in temperature leads to decreased solubility of water in the IL (d) that could then be readily separated to provide the desired fresh water. Upon separation, the recovered IL is recycled for subsequent process turnover of the DSE cycle (e). Given the variable water and salt solubility of many ILs, we were faced with the challenge of identifying a working fluid that would accommodate each phase of this DSE cycle.

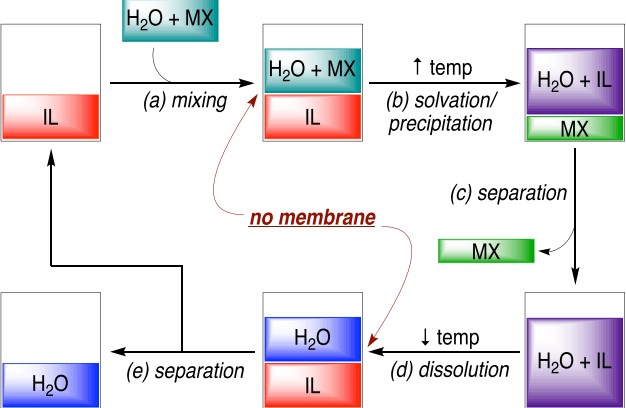

**Fig. 1 Schematic of DSE desalination process employing TSIL.** DSE desalination employing TSILs. (**a**) Saline water mixes with IL; (**b**) Upon heating, IL draws water out of saline while leaving the salts in the water phase, yielding a high salinity brine (MX); (**c**) Brine is separated from the water-in-IL solution; (**d**) A reduction in temperature leads to decreased solubility of water in IL, resulting in dissolution; (**e**) Fresh water is separated from the IL phase, and the IL is recycled for subsequent process turnover of the DSE cycle.

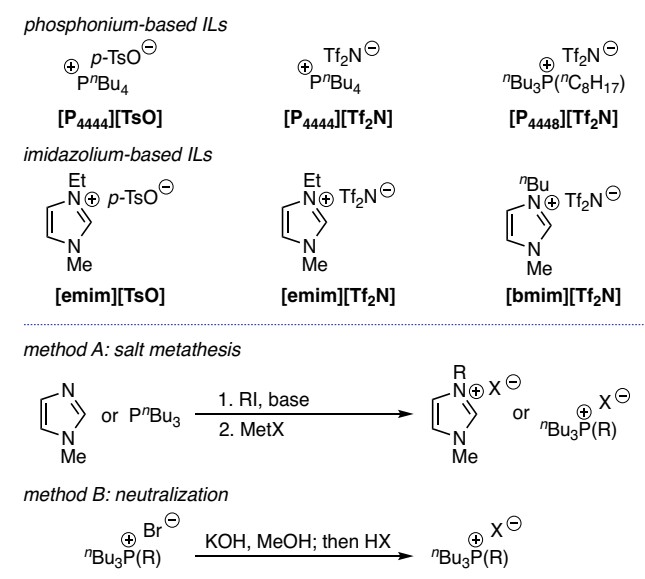

**Fig. 2 Different TSIL candidates investigated for DSE process.** Assembling the TSILs for DSE evaluation. Phosphonium-based and imidazolium-based ILs are studied, and the methods used to synthesize them.

## Results

Inspired by a recent report by Cai and coworkers[18] that demonstrated the feasibility of phosphonium sulfonate ILs as draw solutions for low temperature forward osmosis (FO), we began our study by examining a series of sulfonate anion-based TSILs as DSs for DSE desalination. We initially targeted the p-toluene sulfonate anion salts of tetrabutylphosphonium IL [P$_{4444}$][TsO], and the corresponding 1-ethyl-3-methylimidazolium-based IL [emim][TsO] was also synthesized due to the well-established physical attributes of the imidazolium ILs (Fig. 2). Similarly, the imide anion-based ILs [P$_{4444}$][Tf$_2$N] and [emim][Tf$_2$N] were assembled in an effort to evaluate the relatively non-polar characteristics of the bistriflimide component. To determine the impact of changes in IL hydrophobicity, density and viscosity on DSE performance, we incorporated two different alkyl chain

**Table 1 TSIL DSE desalination performance[a].**

| IL | Ion rejection | Solubility in H$_2$O (ppm) | H$_2$O Yield (%/°C) |
|---|---|---|---|
| [bmim][Tf$_2$N] | 70.5 ± 2.9% | <90 | <0.082 |
| [emim][Tf$_2$N] | 97.5 ± 0.8% | 130–150 | 0.304 ± 0.023 |
| Decanoic acid | 96.9–98.0%[6] | 36–150[6,29] | 0.025 ± 0.002[6] |

[a]ILs found incompatible with DSE are shown in Section 4 in Supplementary Information. [bmim][Tf$_2$N] and decanoic acid group is performed at 75 °C. [emim][Tf$_2$N] is performed at 45, 60, and 75 °C.

lengths in the phosphonium and imidazolium cations to provide the unsymmetrical ILs [P$_{4448}$][Tf$_2$N] and [bmim][Tf$_2$N]. Each targeted IL was readily synthesized using a salt metathesis approach wherein alkylation of either the parent N-methyl imidazole or P$^n$Bu$_3$ with the desired alkyl iodide or bromide followed by anion exchange to incorporate the desired sulfonate or bistriflimide anions. Alternatively, treatment of the tetraalkyl phosphonium bromide with KOH in MeOH followed by exposure to the sulfonic acid or triflimide provided the desired phosphonium ILs. Each IL was obtained in excellent yield and evaluated using ion exchange chromatography and Karl-Fisher titration for residual halide and water content, respectively. Please refer to the "Methods" section for synthesis and characterization of the ILs.

The DS performance in a DSE desalination cycle was evaluated for the assembled ILs, which involved first exposure to water bearing 3.7–5.0 wt% of NaCl (11,000–15,000 ppm of Na$^+$) followed by thorough mixing at elevated temperatures of 45, 60, or 75 °C (see "Methods" section). The mixture was held for ~10 min at each stage to allow for complete phase separation. We note that for the most promising IL ([emim][Tf$_2$N]) identified, a settling time as low as 2-min was found sufficient. The water-enriched IL phase was then removed and allowed to cool to room temperature, which resulted in a second separation of the IL and desalinated water. The amount of residual NaCl and IL in the recovered aqueous phase was then measured using atomic emission spectroscopy and liquid chromatography-mass spectrometry (see Section 3 in the Supplementary Information for details).

Of the characterized ILs, two were identified as potentially viable frameworks for DSE, and their DSE-relevant solubilities are shown in Table 1 with decanoic acid as the comparable standard. Those ILs not shown underwent an undesirable salt metathesis with saline water, exhibited a prohibitively high melting temperature, or exceptionally high viscosity (see Section 4 in Supplementary Information for the list of ILs incompatible with DSE). Curiously, the phosphonium-based ILs either resulted in the instant precipitate formation, presumably a result of a metathesis reaction with NaCl to provide the corresponding [P$_{444R}$][Cl] salt, or exhibited physical properties incompatible with DSE (e.g., melting point >65 °C, high viscosity, etc.). While the imidazolium IL [emim][TsO] also induced a salt metathesis reaction, the bistriflimide [emim][Tf$_2$N] exhibited exceptional ion rejection at >96% with low solubility in water in the range of 130–150 ppm and a surprisingly high-water yield of 0.3%/°C (Table 1). In contrast, [bmim][Tf$_2$N] provided a significantly lower water yield of <0.082%/°C despite a reasonable ion rejection rate of ~70% and favorable solubility in water (<90 ppm).

Given the promising ion rejection rate, water solubility, and water yield exhibited by [emim][Tf$_2$N], we next examined the fresh water recovery of this IL at elevated temperatures. Exposure of [emim][Tf$_2$N] to a 3.7 wt% NaCl saline feed at 45, 60, and 75 °C revealed an average water yield of 0.3 ± 0.023%/°C (Fig. 3a). As is evident, the exceptional water yield displayed by [emim][Tf$_2$N] in comparison to decanoic acid supports our supposition that IL-based DSs constitute a viable class of soft materials toward the development of more efficient DSE desalination process[8].

For saline with NaCl salinity in the range of 3.0–3.8 wt% (11,000–15,000 ppm of Na$^+$), [emim][Tf$_2$N] exhibited outstanding ion rejection rates of 97.0–98.3% (Fig. 3b), bringing the salt content below the drinking water standard of 500 ppm[19]. To test the capacity of [emim][Tf$_2$N] as a DS to treat high salinity water, we also conducted several DSE cycle experiments using 10.8 wt% aqueous NaCl solution (42,600 ppm of Na$^+$) as the feed saline. Gratifyingly, we observed an ion rejection rate of 96.0–96.8% even when [emim][Tf$_2$N] was exposed to high concentrations of aqueous NaCl. These results would indicate that [emim][Tf$_2$N] exhibits promising physical attributes to conduct desalination of water at both moderate and high salinity. Experiments with saturated NaCl used as feed saline is also performed, and the result indicates that we can still effectively perform desalination using DSE with [emim][Tf$_2$N]. The ion rejection rate of the DSE cycle can still reach 96.5% in this case. In the meantime, the fresh water yield indeed drops to 0.157%/°C. The reduction can be understood as that water in higher salinity saline is more thermodynamically stable due to the ion-water electrostatic interaction and more difficult to extract, which was previously examined and also observed for decanoic acid[6]. However, the water yield of IL for saturated saline is still 5.8 times that of decanoic acid treating 3.8% NaCl feed water (0.027%/°C). Additionally, the water yield of [emim][Tf$_2$N] is consistent over a temperature range of 45–75 °C, which suggests that low quality waste heat may be suitable to power this desalination process. All these DSE experiments have been repeated for several cycles by reusing the [emim][Tf$_2$N], and no performance degradation in ion-rejection or water yield is observed. This confirms that ion removal using [emim][Tf$_2$N] in DSE is not due to the IL absorbing NaCl ions from saline water, but caused by IL rejecting these ions. We have performed additional experiments to measure IL concentration in the brine with different salinities. In our experiment, the brine (MX, Fig. 1) salinity is 4% or higher depending on the initial mixing ratio of the saline water and IL. For the MX with 4% salinity, the tested IL concentration is 4 ppm. The IL solubility in the saturated NaCl solution is lower than 4 ppm. As a result, we can conclude that the IL residue in the MX is at a very low level. This is understandable as existing ions in water make the dissolution of additional ions more difficult.

The ion residue in IL, measured by the Na$^+$ concentration, is ~50 ppm. Importantly, we have done DSE for 30 cycles using the same IL and the salt concentration in IL has been steady and the desalination performance has not been degraded. As a result, we believe the ion residue in IL has reached a steady state and will not influence the desalination performance.

Employing molecular dynamics (MD) simulations in combination with thermodynamic integration to calculate the solvation free energies at room temperature for NaCl, water, and [emim][Tf$_2$N], we sought to gain a molecular level understanding of those experimentally observed directional solubilities. In brief, a solute molecule (e.g., NaCl or H$_2$O) is simulated to "gradually appear" into the solution, and free energy needed for this process is calculated as the solvation free energy[20]. The values from our calculation are for solvation from a vacuum state to the solution state, where the vacuum state serves as the common reference for

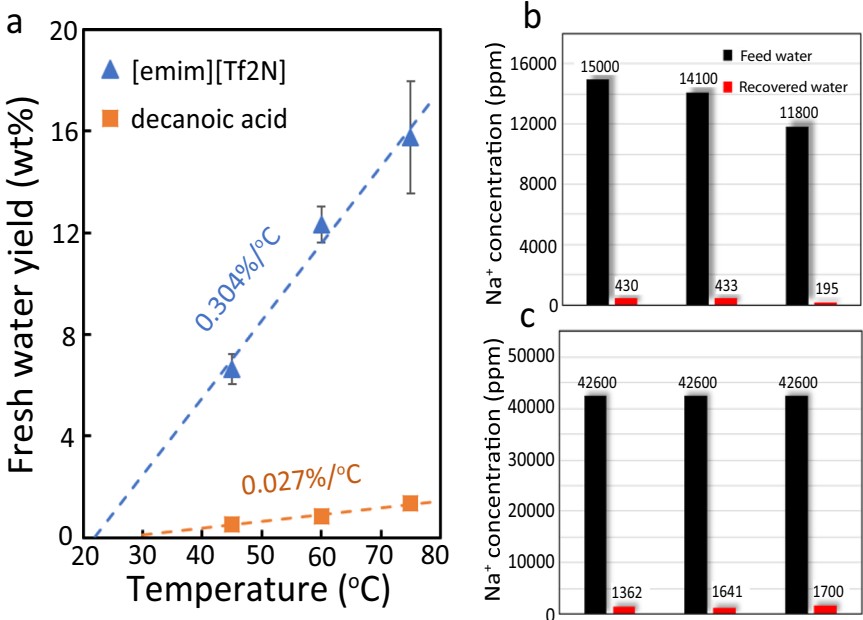

**Fig. 3 Water yield and salt rejection observed in the experiment. a** Water yield and temperature relationship of [emim][Tf$_2$N] and decanoic acid. Error bar is the standard deviation of different tests for each condition. **b** Desalination of moderate salt content water (3.7–5.0 wt%). **c** Desalination of high salt content water (10.8 wt%). Each set of bars in **b** and **c** corresponds to one batch of experiment.

comparing the thermodynamic stability of the solute molecule in different solvents. All calculations are performed at 350 K. Details of the simulations can be found in the "Methods" section. By comparing the solvation free energies, we were able to determine thermodynamic stability in the respective environments for each solute molecule, which ultimately aided our understanding of the variable solvation tendencies. The calculated solvation free energy of NaCl in [emim][Tf$_2$N] is −677.7 kJ/mol, which is in contrast to −699.5 kJ/mol found for NaCl in water. This indicates that the NaCl salt favors solvation in the aqueous media over the corresponding IL phase, and rationalizes the observed ion rejection capability of [emim][Tf$_2$N] in the DSE process. Similarly, the solvation free energy of water in [emim][Tf$_2$N] of −26.5 kJ/mol is lower than that of water in water (−22.9 kJ/mol), which is consistent with the observed propensity for water to dissolve into the IL. Additionally, the calculated solvation free energy of −38.6 kJ/mol for [emim][Tf$_2$N] in itself is significantly less than that of [emim][Tf$_2$N] in water (−23.3 kJ/mol), which suggests that it is thermodynamically unfavorable for the IL to dissolve in water. We note that the above two cases related to [emim][Tf$_2$N] solvation used the state of a [emim][Tf$_2$N] ionic pair in vacuum as the reference level. Overall, these calculations are consistent with our experimental observations, revealing that [emim][Tf$_2$N] displays favorable DS thermodynamic properties of water insolubility while concurrently capable of solvating water molecules and rejecting salt ions. The above simulations used the TIP4P as the water model. We have also used the TIP3P water model and found the same solvation tendencies (see Supplementary Information, Table S5).

Besides the calculation of solvation free energy, we also ran a simulation of 3.7% NaCl water solution in contact with [emim][Tf$_2$N] at 350 K for a duration of 30 ns (see Supplementary Information, Section 7 for simulation details). Figure 4a shows snapshots of the simulation of the ternary system. Throughout the simulation, almost all Na$^+$ and Cl$^−$ remained in the water phase with only two of them appeared to have diffusing into the [emim][Tf$_2$N] phase. Even for those diffused into [emim][Tf$_2$N], they are surrounded by water molecules in the IL. A large number of water molecules diffused into the [emim][Tf$_2$N] phase, but only a limited number of [emim]$^+$ and [Tf$_2$N]$^−$ ions got into the water phase. Figure 4b shows the density profiles of water and IL at different times corresponding to the snapshots in Fig. 4a. It is apparent that water diffusion into IL is more significant than IL diffusion into water. These phenomena generally agree with the solvation-free energy calculation results and experimental observations. We have also analyzed the bonding nature between water and IL molecules and found hydrogen bonds exist between water and the [Tf$_2$N]$^−$ ions (Fig. 4c). The hydrogen bonds are formed between the water molecules and the polar sulfonyl groups of the [Tf$_2$N]$^−$ ions (inset in Fig. 4c), and the number of hydrogen bonds grows as more water molecules dissolved into IL.

The ability to operate a DSE process at the relatively mild temperatures shown with [emim][Tf$_2$N] (e.g., 45 °C) constitutes a significant advantage wherein waste heat is a viable source of operating power. Exergy is the maximum amount of extractable work from a heat source with Carnot efficiency, and is a useful measure of the overall system efficiency when comparing different desalination technologies[6,21]. As shown in Fig. 5 (see "Methods" section), exergy consumptions of DSE with [emim][Tf$_2$N] and decanoic acid both increase with increasing heat source temperature. A DSE process operated with [emim][Tf$_2$N] has an exergy cost of 2.4 kWh/m$^3$ at 45 °C and 5.9 kWh/m$^3$ at 75 °C. The pumping power requirement is 0.31 kWh/m$^3$ at 45 °C, and this value will be smaller at higher temperature, according to another study[22] (details of pumping power calculation in Section 9 of Supplementary Information). Comparing the exergy consumptions of DSE processes reveals a staggering reduction in exergy cost by 70% at 45 °C when [emim][Tf$_2$N] is employed as the DS in contrast to decanoic acid[6]. Additionally, this observed exergy penalty reduction increases to 89% at 75 °C. Exergy consumption of a state-of-the-art thermal desalination technology—multistage flash (MSF)[4,23–27] is also provided for comparison. Besides the advantage of utilizing lower temperature (45–75 °C) heat sources compared to MSF (>85 °C), the exergy consumption of DSE using [emim][Tf$_2$N] is significantly lower than that of the MSF. While decanoic acid exhibits comparable

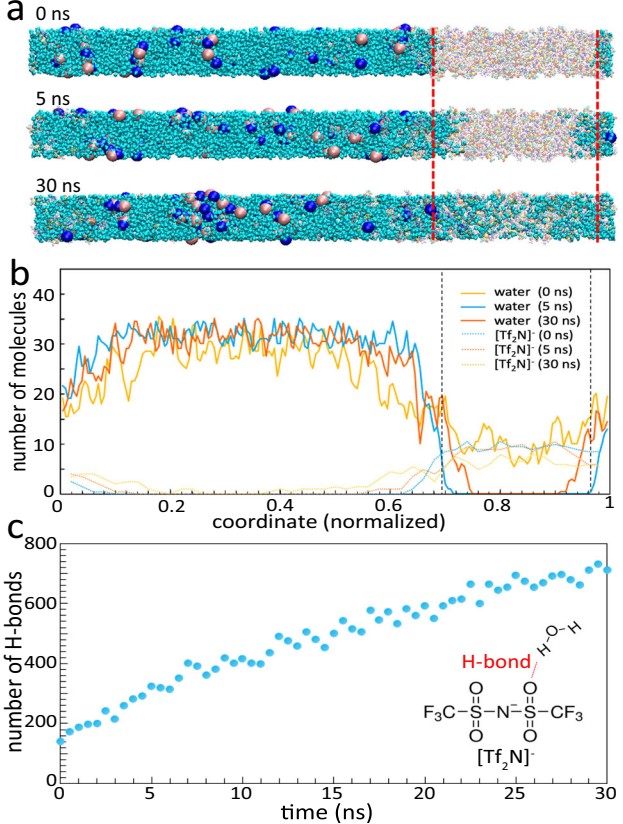

**Fig. 4 MD simulation of the interface between IL and NaCl water solution (3.7%). a** Snapshots of MD simulations at different times, where the blue block is water, red block is IL and large pink and dark blue spheres are respectively Cl⁻ and Na⁺ ions. **b** Density profiles of water and $[Tf_2N]^-$ of the IL at different times corresponding to the snapshots in **a**. We note the $[emim]^+$ shows similar profile as the $[Tf_2N]^-$ and it is not plotted for clarity of the figure. **c** Number of hydrogen bonds (H-bonds) between $[Tf_2N]^-$ and water molecules as a function of time.

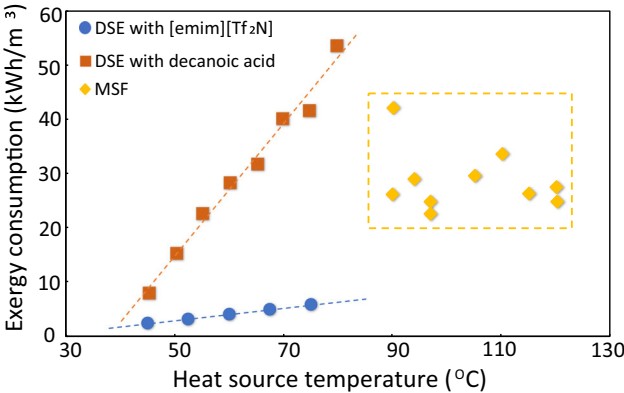

**Fig. 5 Exergy consumption as a function of heat source temperature.** Comparison of exergy consumption of directional solvent extraction (DSE) using $[emim][Tf_2N]$ and decanoic acid and multi-stage flash (MSF). Propagated from the data uncertainty from Fig. 3, the calculated uncertainty of the blue data points ranges from 20% (at 45 °C) to 17% (75 °C).

performance advantages over MSF, these are only realized at operating temperatures below ~55 °C (Fig. 5). We note that ref. 3 reported that an energy consumption as low as 2 KWh/m³ from new large-scale RO plants might be achieved. Another study[28]

showed that in the state-of-the-art RO plants, the energy cost related to the RO process alone, excluding those from processes like pre-treatment and water delivery, is 2.54 kWh/m³. As a result, DSE has similar energy cost as RO when operating at low temperature, let alone its unique capability of harvesting low temperature waste heat. Moreover, DSE will be suitable for small scale application, especially for low-resource settings, where the energy cost for small-scale RO is high (up to 17 kWh/m³)[4], largely due to the lack of centralized infrastructures for mechanical energy recovery.

While residual $[emim][Tf_2N]$ in the recovered water should be low (130–150 ppm), there is no established toxicity standard for this IL in drinking water. To completely remove and recover all IL from the produced water, we can use nanofiltration (NF) as implemented by Cai et al.[18] and Elimelech et al.[3] for their IL-FO desalination for similar residue removal purposes. It will also eliminate the loss of IL during operation. The exergy cost for the NF process is a mere 0.011 KWh/m³ given the low osmotic pressure associated with a minimal concentration of residual IL, thereby avoiding a significant energy penalty (see Section 8 in Supplementary Information for calculation details).

## Discussion

The efficacy of IL-based DSs appear to be modulated dramatically by structural perturbations of the IL. The optimal IL discovered being $[emim][Tf_2N]$ contains small alkyl chains adorned on the imidazolium cation and a hydrophobic resonance stabilized bis-triflimide anion. Conversely, the ILs which failed to provide DSE behavior contain structural features altering hydrophobicity, size, and charge dispersion of the anion and cation. As in the case of $[emim][TsO]$, which contains the larger $[TsO]^-$ anion in comparison to $[emim][Tf_2N]$, a precipitate was observed upon subjection to a DSE cycle, presumably a mixture of $[emim][Cl]$ and $[Na][TsO]$. We attribute the failure in DSE behavior to $[emim][TsO]$, forming a less intimate ion pair, allowing for salt metathesis reactions to become more favorable. In addition, both components of $[emim][TsO]$ contain aromatic systems, which may lead to poor water solubility. The $[bmim][Tf_2N]$ IL, while providing moderate DSE behavior, failed to reject NaCl (~70%) as efficiently as $[emim][Tf_2N]$ or decanoic acid and has lower solubility in water (<90 ppm) as well as water yield (0.082 %/°C), which we attest to the lengthened aliphatic chain. The $[bmim]^+$ cation has an increased hydrophobicity due to increased chain length as well as a weaker cation–anion interaction as longer aliphatic chain lengths present larger steric interaction with the $[Tf_2N]^-$ anion. Therefore, a stronger interaction may be achieved between $[bmim][Cl]$ and $[Na][Tf_2N]$ due to the relative size differences yielding more intimate ion pairs sequestering more NaCl. As a result, the salt rejection rate is only around 70% according to our tests. It, however, might be used for treating low salinity water (<1600 ppm), where the 70% ion rejection rate can bring the salinity down to meet the drinking water standard of 500 ppm. The increased hydrophobicity of the $[bmim]^+$ cation compared to the $[emim]^+$ cation can also explain its comparatively lower water yield. We have performed MD simulations to calculate the free energy of water in $[bmim][Tf_2N]$, and the result (−24.3 kJ/mol) is indeed higher than that of water in $[emim][Tf_2N]$ (−26.5 kJ/mol).

In comparison to the imidazolium-based cations whose positive charge is resonated into the aromatic system, the tetraalkyl phosphonium cations have a point charge dispersed over a smaller area (See Section 1 in Supplementary Information for tetraalkyl phosphonium-based ILs used in this study). This difference in polarization alters ion-pair strength, which may be responsible for the decrease in the desired DS behavior of

tetralkyl phosphonium ILs, where larger charge separation weakens the interior interactions of the IL, allowing large amounts of sodium chloride to penetrate into the IL phase leading to undesired metathesis reactions. In addition to size and charge distribution, the viscosity of the IL affects the DSE behavior. For example, the ion rejection rate is found to be very low for $[P_{4448}][Tf_2N]$, (The concentration of recovered water is even higher than the feed water.) whose viscosity is 0.433 Pa s, which is very high (See Section 4 in Supplementary Information for viscosity of the ILs). Furthermore, high viscosity ILs are not optimal for practical applications as a large pumping power would be required to perform a DSE cycle.

In summary, we have demonstrated that DSE can effectively extract fresh water from saline sources, even saturated saline water, by using $[emim][Tf_2N]$. The DSE technique, which can utilize the low temperature heat, can potentially result in low resource setting applications. By analyzing the results of several tested ILs and their DSE behavior, we have also rationalized the chemistry–property relation, which might be helpful for further identification of task specific DS ILs. The work is also expected to further stimulate research around the DSE technology, especially the exploration of even higher-performance DSs. To move DSE closer to practical deployment, mass production of the identified ILs, process optimization, thermal system design with the potential integration of waste heat or renewable energy are topics worth additional research or engineering.

## Methods

**IL synthesis and characterization**. Synthesis and characterization details of the ILs are included in Section 1 in the Supplementary Information.

**DSE experiments**. Detailed procedure and experimental parameters of the DSE process to screen candidate ILs are included in Section 2 in the Supplementary Information.

**MD simulations**. Details of MD simulations and free energy calculation using thermodynamic integration are discussed in Sections 5 and 6 in Supplementary Information.

**Energy calculation**. Details of energy and exergy calculations are included in Section 7 of the Supplementary Information.

## Data availability

All datasets generated during and/or analyzed during the current study are available from the corresponding author on reasonable request.

## Code availability

All code and input files used in the current study are available from the corresponding author on reasonable request.

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

## Acknowledgements

This research was supported by National Science Foundation (1510826, CHE 1665440 and 1956170, and CBET 2031431). We would like to thank the Center for Environmental Science and Technology (CEST) and Mass Spectrometry & Proteomics Facility for facilitating measurement of the NaCl and IL concentrations. We also thank the ND Energy Materials Characterization Facility (MCF) for the measurement of liquid viscosity. The MCF is funded by the Sustainable Energy Initiative (SEI), which is part of the Center for Sustainable Energy at Notre Dame (ND Energy). The simulation is partially supported by the Center for Research Computing at the University of Notre Dame and the NSF through XSEDE computing resources provided by TACC Stampede-2 under Grant No. TG-CTS100078.

## Author contributions

B.L.A. and T.L. designed the experiments. J.G. and Y.W. carried out desalination experiments and molecular simulation. Z.D.T. synthesized and characterized the ionic liquids. All authors discussed the results and wrote the manuscript.

## Competing interests

The authors declare no competing interests.
