## [Peer Review File · Nature Communications]

REVIEWER COMMENTS

Reviewer #1 (Remarks to the Author):

The manuscript by Guo et al. reports the results of joint experimental/simulation investigation of several ionic liquids as potential candidates for directional solvents (DS) in water desalination process using directional solvent extraction (DSE). DSE is an attractive approach alternative to conventional membrane separation and relies on the interplay of thermodynamic interactions between ions, water, and solvent. Authors point out that currently best performing DS has a very low water yield. The manuscript demonstrates that utilization of [emim][Tf2N] can increase the water yield by factor of 10 and provide a very high ion rejection fraction. Therefore this communication points out to a new direction in the search of promising directional solvents for this technologically and scientifically interesting process.

Overall, the manuscript is well-organized, clearly written, and will be of interest to a broad scientific community. However, I have several comments that authors need to address before I can recommend this manuscript for publication:

1) While finding a DS that appears to perform substantially better than other solvents is important, it is as much important to understand why this particular ionic liquid [emim][Tf2N] showed such characteristics. Experimental data show that by slightly changing IL cation from emim to bmim the performance changes significantly. Yet, no modeling has been done to calculate the free energies of solvation for [bmim][TF2N] and to compare them with the data reported for [emim][Tf2N]. Such direct comparison would help to support somewhat speculative statements in the Discussion section as well as to demonstrate that proposed modeling algorithm can be used to identify promising DS candidates.

2) How accurate are the force fields employed in molecular dynamics simulations in their ability to capture thermodynamics of investigated mixtures? It is well-known that the solubility in ionic systems can be quite sensitive to the choice of the force field. In this study, authors use a non-polarizable force field, which for ionic system can lead to noticeable overestimation of binding energies, and the TIP3P water model, which also has some drawbacks in capturing thermodynamics of water. Taking this into account, a direct comparison of solvation free energies obtained in this work for Na⁺ in [emim][Tf2N] with that reported in ref 22 for Na⁺ in water using a completely different force field is somewhat meaningless and misleading. If authors want to compare solvation energetics they need to do it with the same consistent force field. Therefore, I strongly recommend to do additional simulations and to get all discussed free energies using consistent model. (Same issue is for comparison of Cl⁻ free energies with those reported in ref. 7).

3) In the discussion of exergy and estimation of pumping power requirements it is not clear what flow rate or flow velocity is assumed in those calculations.

Reviewer #2 (Remarks to the Author):

The authors report an experimental and computational study for the usage of ionic liquids as directional solvents in desalination processes. This is an extremely interesting up-to-date research topic, currently under very active scrutiny. The detection of new media for this purpose is a need, and work in this direction must be valued accordingly. The present paper contains a very valuable contribution performed with adequate computational techniques and appropriate protocols, and it provides detection of the optimal candidate among several studied phosphonium sulphonate ionic liquids (Emim)[Tf₂N] for this task. Moreover, the authors aim to provide a microscopic picture of the solvation free energies of mixtures of this ionic liquids with water and NaCl together with exergetic analysis of the process. The conclusions appear to me as essentially correct, and I have not found any serious flaw in it. The clarity and length of the manuscript are also appropriate, and it is adequately referenced. The number and quality of figures are acceptable. The paper is well written and the quality of the English employed in the text is very good, although some minor mistakes that might very well be corrected in the proofs were detected.

I recommend it is ACCEPTED for publication in Nature Communications, but before the final acceptance of the ms. I would like the authors to be a little bit more specific with the solvation details in their systems, which they can easily do given that they already have the simulation trajectories. Could they include, for the interest of readers, some details about the distribution functions of the different species in the ternary mixtures (NaCl+water+Ionic liquid)? Have they tried different concentrations of the salt and water in the studied mixtures? At the same time that it would help understanding the energetics of solvation, this would provide very valuable information about how the solutions of the salt ions is actually done in the mixtures, and would be very convenient for future contributions and analysis of these systems.

Reviewer #3 (Remarks to the Author):

The work by Guo et al using ionic liquid (IL) as the directional solvent (DS) in DSE is highly interesting showing 10 times higher water yield than that from the literature reported DS. The manuscript is also presented with reasonable clarity. However, significant improvement should be made on the manuscript before being considered for publication. Below are specific comments and suggestions on how this may be done.

What would be a desirable water yield target for DSE? The overall water yield is still very low, only ~6.5% at 45°C and ~16% even at 75°C (see figure 3). How do these values fare in comparison to other desalination technologies? Also, there should be a quantitative analysis of IL concentration in the brine MX (after Step (c) in Figure 1), as well as the concentration of salt left in the IL after Step (e) in figure 1. These important data are missing in the current manuscript.

It is impressive that there is only a small reduction in salt rejection rate (less than 2 %) when higher salinity of water of 42,600 ppm of sodium was used for the DSE test. However, the authors should report the corresponding fresh water yield at such a high salinity as well. The authors should at least discuss briefly whether there is an upper limit of salinity beyond which the ionic liquid-DSE becomes

impractical.

There is some puzzle concerning the MD calculation. Is there any rational explanation why the mentioned IL has such a large positive solvation free energy in itself? Besides, are all those free energy values obtained at the same temperature, or at different temperatures, knowing that different temperatures were used for water extraction into IL and subsequent water release from the IL? This is unclear in the manuscript or the supplementary information.

The statement "In summary, we have demonstrated that DSE can effectively remove salt ions from water by using [emim][Tf2N]" is inaccurate. Strictly speaking, DSE did not remove salt from water, but rather remove water from aqueous saline solution. Salt extraction and water extraction are in fact two fundamentally different processes in the field of desalination.

Finally, it is suggested that the authors provide a concise overall discussion on DSE technology not only highlighting future opportunities but also the remaining challenges and limitations of the start-of-the-art development including the work presented in this manuscript.

Dear Editor & Reviewers,

We appreciate the time and effort you put to the editorial and reviewing processes regarding our manuscript, “Ionic Liquid Enables Highly Efficient Low Temperature Desalination by Directional Solvent Extraction”, by Jiayi Guo, Zachary Tucker, Yu Wang, Brandon Ashfeld and Tengfei Luo. We have carefully studied the reviewers’ comments and included our point-by-point responses below. We have also modified the manuscript accordingly, and the changes are highlighted.

Reviewer 1

The manuscript by Guo et al. reports the results of joint experimental/simulation investigation of several ionic liquids as potential candidates for directional solvents (DS) in water desalination process using directional solvent extraction (DSE). DSE is an attractive approach alternative to conventional membrane separation and relies on the interplay of thermodynamic interactions between ions, water, and solvent. Authors point out that currently best performing DS has a very low water yield. The manuscript demonstrates that utilization of [emim][Tf₂N] can increase the water yield by factor of 10 and provide a very high ion rejection fraction. Therefore this communication points out to a new direction in the search of promising directional solvents for this technologically and scientifically interesting process.

Overall, the manuscript is well-organized, clearly written, and will be of interest to a broad scientific community. However, I have several comments that authors need to address before I can recommend this manuscript for publication:

1) While finding a DS that appears to perform substantially better than other solvents is important, it is as much important to understand why this particular ionic liquid [emim][Tf₂N] showed such characteristics. Experimental data show that by slightly changing IL cation from emim to bmim the performance changes significantly. Yet, no modeling has been done to calculate the free energies of solvation for [bmim][Tf₂N] and to compare them with the data reported for [emim][Tf₂N]. Such direct comparison would help to support somewhat speculative statements in the Discussion section as well as to demonstrate that proposed modeling algorithm can be used to identify promising DS candidates.

Reply: We thank the reviewer for bringing up this great suggestion. We have now performed free energy calculations for [bmim][Tf₂N] as a comparison to [emim][Tf₂N]. We note that based on the reviewer’s 2nd comment below, the more accurate TIP4P model for water is used to calculate the solvation energy in all our new simulations. As shown in the table below, we can see that water indeed has lower free energy when dissolved in [bmim][Tf₂N] compared to that in the bulk water phase, meaning water will likely dissolve in [bmim][Tf₂N]. However, the [bmim][Tf₂N] environment is not as favorable for water as [emim][Tf₂N], suggesting that water will have lower solubility in [bmim][Tf₂N] than in [emim][Tf₂N]. These are consistent with our experimental observations. We note that [bmim][Tf₂N] has a longer alkyl side chain (4 carbon segments) attached to the imidazolium than [emim][Tf₂N] (2 carbon segments). These side chains are hydrophobic, and thus the longer side chain in [bmim][Tf₂N] understandably leads to lower water solubility.

Table: Free energy of solvating a water molecule (TIP4P) in different solvents (77°C)

solute	solvent	free energy of solvation (kJ/mol)
TIP4P	TIP4P	-22.9
TIP4P	[bmim][Tf ₂ N]	-24.3
TIP4P	[emim][Tf ₂ N]	-26.5

We have added the following discussion to the Discussion section (page 9 of revised manuscript) to include the above discussion.

“The increased hydrophobicity of the [bmim]⁺ cation compared to the [emim]⁺ cation can also explain its comparatively lower water yield. We have performed MD simulations to calculate the free energy of water in [bmim][Tf₂N], and the result (-24.3 kJ/mol) is indeed higher than that of water in [emim][Tf₂N] (-26.5 kJ/mol).”

2) How accurate are the force fields employed in molecular dynamics simulations in their ability to capture thermodynamics of investigated mixtures? It is well-known that the solubility in ionic systems can be quite sensitive to the choice of the force field. In this study, authors use a non-polarizable force field, which for ionic system can lead to noticeable overestimation of binding energies, and the TIP3P water model, which also has some drawbacks in capturing thermodynamics of water. Taking this into account, a direct comparison of solvation free energies obtained in this work for Na⁺ in [emim][Tf₂N] with that reported in ref 22 for Na⁺ in water using a completely different force field is somewhat meaningless and misleading. If authors want to compare solvation energetics, they need to do it with the same consistent force field. Therefore, I strongly recommend to do additional simulations and to get all discussed free energies using consistent model. (Same issue is for comparison of Cl⁻ free energies with those reported in ref. 7).

Reply: We thank the reviewer for this comment, which we agree. Besides the TIP3P model, we have used the more sophisticated TIP4P model¹ for water to calculate all the relevant free energies again. In addition, we have performed a thorough re-visit of all previous calculations and adjusted some results due to better molecular system relaxation. Now, all the calculated values are from the same type of simulations with the same type of force field. The table below summarizes all results. Since using TIP3P and TIP4P water models yield similar solubility tendencies, which all agree with the experimental observation, we believe the simulations using these force fields are informative and support the correct physics.

Table: Free energy of solvation list (77 °C)

solute	solvent	free energy of solvation (kJ/mol)
TIP4P	TIP4P	-22.9
TIP4P	[emim][Tf ₂ N]	-26.5
[emim][Tf ₂ N]	TIP4P	126.2
[emim][Tf ₂ N]	[emim][Tf ₂ N]	110.9
NaCl	TIP4P	-699.5
NaCl	[emim][Tf ₂ N]	-677.7
TIP3P	TIP3P	-23.9
TIP3P	[emim][Tf ₂ N]	-24.0
NaCl	TIP3P	-709.6
[emim][Tf ₂ N]	TIP3P	134.4

From the table, we can conclude that the free energy results agree with our experimental observations, i.e., [emim][Tf₂N] will dissolve water and at the same time, [emim][Tf₂N] is not likely to dissolve in the water phase. In addition, NaCl will stay in the water phase instead of dissolving into the [emim][Tf₂N] phase. We have updated all the free energy results in the manuscript (page 6 of revised

manuscript) and included those using the TIP3P water model in the Supporting Information (see modification on page 23 of the SI). We have also copied the edited part here for convenience:

“The calculated solvation free energy of NaCl in [emim][Tf₂N] is -677.7 kJ/mol, which is in contrast to -699.5 kJ/mol found for NaCl in water. This indicates that the NaCl salt favors solvation in the aqueous media over the corresponding IL phase, and rationalizes the observed ion rejection capability of [emim][Tf₂N] in the DSE process. Similarly, the solvation free energy of water in [emim][Tf₂N] of -26.5 kJ/mol is lower than that of water in water (-22.9 kJ/mol), which is consistent with the observed propensity for water to dissolve into the IL. Additionally, the calculated solvation free energy of 110.9 kJ/mol for [emim][Tf₂N] in itself is significantly less than that of [emim][Tf₂N] in water (126.2 kJ/mol), which suggests that it is thermodynamically unfavorable for the IL to dissolve in water. We note that while the above two cases related to [emim][Tf₂N] solvation yield positive free energy values, it does not mean that [emim][Tf₂N] are not stable in these environments, since the free energy of a [emim][Tf₂N] in vacuum, which is the reference level, is 149.5 kJ/mol. Overall, these calculations are consistent with our experimental observations, revealing that [emim][Tf₂N] displays favorable DS thermodynamic properties of water insolubility while concurrently capable of solvating water molecules and rejecting salt ions. The above simulations used the TIP4P as the water model. We have also used the TIP3P water model and found the same solvation tendencies (see Supporting Information, Table S5).”

3) In the discussion of exergy and estimation of pumping power requirements it is not clear what flow rate or flow velocity is assumed in those calculations.

Reply: We thank the reviewer for this comment. All our calculations were based on a production rate of 1 kg/s fresh water, but the total fluid (IL+water) flow rate used for the calculations depended on the working temperature, since the absolute yield is linear to temperature difference. The pump power would be dominantly devoted for IL pumping due to its much larger volume than water. If the system operates between 20 °C and 45 °C, the required IL flow rate is 13.3 kg/s and the resultant flow velocity is 0.28 m/s (as indicated in the SI, the pipe inner diameter is 0.2 m). We have added this information to the Supporting Information on page 25.

Reviewer 2

The authors report an experimental and computational study of the usage of ionic liquids as directional solvents in desalination processes. This is an extremely interesting up-to-date research topic, currently under very active scrutiny. The detection of new media for this purpose is a need, and work in this direction must be valued accordingly. The present paper contains a very valuable contribution performed with adequate computational techniques and appropriate protocols, and it provides detection of the optimal candidate among several studied phosphonium sulphonate ionic liquids (Emim)[Tf₂N] for this task. Moreover, the authors aim to provide a microscopic picture of the solvation free energies of mixtures of this ionic liquids with water and NaCl together with exergetic analysis of the process. The conclusions appear to me as essentially correct, and I have not found any serious flaw in it. The clarity and length of the manuscript are also appropriate, and it is adequately referenced. The number and quality of figures are acceptable. The paper is well written and the quality of the English employed in the text is very good, although some minor mistakes that might very well be corrected in the proofs were detected.

I recommend it is ACCEPTED for publication in Nature Communications, but before the final acceptance of the ms. I would like the authors to be a little bit more specific with the solvation details in their systems, which they can easily do given that they already have the simulation trajectories. Could they include, for the interest of readers, some details about the distribution functions of the different species in the ternary mixtures (NaCl+water+Ionic liquid)? Have they tried different concentrations of the salt and water in the studied mixtures? At the same time that it would help understanding the energetics of solvation, this would provide very valuable information about how the solutions of the salt ions is actually done in the mixtures, and would be very convenient for future contributions and analysis of these systems.

Reply: We thank the reviewer for these great comments. First, we have now added the simulation results of the ternary mixture, and the result is updated in the manuscript (page 7). We have also copied the addition below:

“Besides the calculation of solvation free energy, we also run a simulation of 3.7% NaCl water solution in contact with [emim][Tf₂N] at 350 K for a duration of 30 ns (see Supporting Information, Section 7 for simulation details). Figure 4a shows snapshots of the simulation of the ternary system. Throughout the simulation, almost all Na⁺ and Cl⁻ remained in the water phase with only two of them appeared to have diffusing into the [emim][Tf₂N] phase. Even for those diffused into [emim][Tf₂N], they are surrounded by water molecules in the IL. A large number of water molecules diffused into the [emim][Tf₂N] phase, but only limited number of [emim]⁺ and [Tf₂N]⁻ ions got into the water phase. Figure 4b shows the density profiles of water and IL at different times corresponding to the snapshots in Fig. 4a. It is apparent that water diffusion into IL is much more significant than IL diffusion into water. These phenomena generally agree with the solvation free energy calculation results and experimental observations. We have also analyzed the bonding nature between water and IL molecules and found hydrogen bonds exist between water and the [Tf₂N]⁻ ions (Fig. 4c). The hydrogen bonds are formed between the water molecules and the polar sulfonyl groups of the [Tf₂N]⁻ ions (inset in Fig. 4c), and the number of hydrogen bonds grows as more water molecules dissolved into IL.

Figure 4. MD simulation of the interface between IL and NaCl water solution (3.7%). (a) Snapshots of MD simulations at different times, where the blue block is water, red block is IL and large pink and dark blue spheres are respectively Cl⁻ and Na⁺ ions. (b) Density profiles of water and [Tf₂N]⁻ of the IL at different times corresponding to the snapshots in panel (a). (c) Number of hydrogen bonds (H-bonds) between [Tf₂N]⁻ and water molecules as a function of time.”

Second, in addition to the originally tested salinities of 3.8% and 10.5% NaCl, we have added 26.5% NaCl solution (i.e., saturated) as the feed water. We have observed that as the feed water salinity increases, the water yield using [emim][Tf₂N] decreases as expected. This is because higher salinity makes the solution a more favorable environment for water molecules, which also makes it harder for the IL to extract them. These are consistent with our previous studies on decanoic acid as a directional solvent.^{2,3} These new results are included in the revised manuscript (page 5). We have also copied the edited parts below for convenience:

“Experiments with saturated NaCl used as feed saline is also performed, and the result indicates that we can still effectively perform desalination using DSE with [emim][Tf₂N]. The ion rejection rate of the DSE cycle can still reach 96.5% in this case. In the meantime, the fresh water yield indeed drops to 0.157%/°C. The reduction can be understood as that water in higher salinity saline is more thermodynamically stable due to the ion-water electrostatic interaction and more difficult to extract, which we previously examined and also observed for decanoic acid.⁶ However, the water yield of IL for saturated saline is still 5.8 times of that of decanoic acid treating 3.8% NaCl feed water (0.027%/°C).”

Reviewer 3

The work by Guo et al using ionic liquid (IL) as the directional solvent (DS) in DSE is highly interesting showing 10 times higher water yield than that from the literature reported DS. The manuscript is also presented with reasonable clarity. However, significant improvement should be made on the manuscript before being considered for publication. Below are specific comments and suggestions on how this may be done.

1) What would be a desirable water yield target for DSE? The overall water yield is still very low, only ~6.5% at 45°C and ~16% even at 75°C (see figure 3). How do these values fare in comparison to other desalination technologies? Also, there should be a quantitative analysis of IL concentration in the brine MX (after Step (c) in Figure 1), as well as the concentration of salt left in the IL after Step (e) in figure 1. These important data are missing in the current manuscript.

Reply: We thank the reviewer for these comments. DSE is a very unique desalination technique invented a few years ago. The only mature solvent for DSE is decanoic acid, and thus it was the only material we compared to in this study. The new IL is actually much better than decanoic acid. Due to the distinct desalination mechanism compared to other technologies (e.g., distillation and reverse osmosis), we compared their energy and exergy consumption to highlight the uniqueness of the DSE when using our newly discovered IL as the DS (see pages 7 and 8 of the manuscript for these discussions). The new IL solvent actually make DSE to be much more competitive than state-of-the-art thermal distillation technology as shown in Fig. 5 of the manuscript. The discussion on page 8 of the manuscript also indicate that DSE using the new IL is similar in energy consumption compared to RO.

We have now added quantitative analysis of IL concentration in the brine and the salt concentration in IL. The results and the discussion are added to page 5 of the revised manuscript. They are also copied below:

“We have performed additional experiments to measure IL concentration in the brine with different salinities. In our experiment, the brine (MX, Fig. 1) salinity is 4% or higher depending on the initial mixing ratio of the saline water and IL. For the MX with 4% salinity, the tested IL concentration is 4 ppm. The IL solubility in the saturated NaCl solution is further lower than 4 ppm. As a result, we can conclude that the IL residue in the MX is at a very low level. This is understandable as existing ions in water make the dissolution of additional ions more difficult.

The ion residue in IL, measured by the Na⁺ concentration, is ~50 ppm. Importantly, we have done DSE for 30 cycles using the same IL and the salt concentration in IL has been steady and the desalination performance has not been degraded. As a result, we believe the ion residue in IL has reached a steady state and will not influence the desalination performance.”

2) It is impressive that there is only a small reduction in salt rejection rate (less than 2 %) when higher salinity of water of 42,600 ppm of sodium was used for the DSE test. However, the authors should report the corresponding fresh water yield at such a high salinity as well. The authors should at least discuss briefly whether there is an upper limit of salinity beyond which the ionic liquid-DSE becomes impractical.

Reply: We thank the reviewer for this great suggestion. We performed additional experiments to address this comment. From the experiment results, we can see in the case where saturated NaCl is used as the feed saline, we can still effectively perform desalination using DSE with the IL. The ion rejection rate of the DSE cycle can still reach 96.5% in this case. In the meantime, the fresh water yield indeed drop to 0.157%/°C. The reduction can be understood as that water in higher salinity saline is more thermodynamically stable due to the ion-water electrostatic interaction and more difficult to extract, which we previously examined and also observed for decanoic acid². However, the water yield of IL for saturated saline is still 5.8 times of that of decanoic acid treating 3.8% NaCl feed water (0.027%/°C). As a result, we can conclude that [emim][Tf₂N] is a good candidate DS that can work on a wide range of salinity water, and is especially promising for hypersaline desalination for the potential “zero liquid

discharge". A detailed comparison of the results with saline of different salinities is included in the table below.

Table: Comparison of results from [emim][Tf₂N] DSE cycle treating saline with different salinities.

Feeding concentration	3.8% NaCl	26.5% NaCl (Saturated)
Corresponding Na ⁺ concentration (ppm)	14000	95000
Ion rejection rate	97.5%	96.8%
Na ⁺ concentration in recovered water (ppm)	430	3000
Na ⁺ residual in [emim][Tf ₂ N] (ppm)	50	1095
Water yield (%/°C)	0.304	0.157

These discussions are added to page 5 of revised manuscript and copied below:

“Experiments with saturated NaCl used as feed saline is also performed, and the result indicates that we can still effectively perform desalination using DSE with [emim][Tf₂N]. The ion rejection rate of the DSE cycle can still reach 96.5% in this case. In the meantime, the fresh water yield indeed drops to 0.157%/°C. The reduction can be understood as that water in higher salinity saline is more thermodynamically stable due to the ion-water electrostatic interaction and more difficult to extract, which we previously examined and also observed for decanoic acid.⁶ However, the water yield of IL for saturated saline is still 5.8 times of that of decanoic acid treating 3.8% NaCl feed water (0.027%/°C).”

3) There is some puzzle concerning the MD calculation. Is there any rational explanation why the mentioned IL has such a large positive solvation free energy in itself? Besides, are all those free energy values obtained at the same temperature, or at different temperatures, knowing that different temperatures were used for water extraction into IL and subsequent water release from the IL? This is unclear in the manuscript or the supplementary information.

Reply: We thank the reviewer for these questions. First, we should mention all calculations are performed in the same temperature of 77 °C (350 K), which is close to the highest temperature used in the experiment. Higher temperature helps the convergence of the simulation since dynamics is faster, but the solvation tendencies extracted from these simulations should also be applicable to lower temperatures. We have now mentioned this in the revision. IL indeed turns out to have a positive solvation free energy in itself, but the value is smaller than that of IL in water. As can be inferred from the Method section, the IL molecule in vacuum is the reference level. We have calculated the free energy of a [emim][Tf₂N] in vacuum, which is the reference level, is 149.5 kJ/mol, which is higher than the dissolved cases, meaning the dissolved cases are more stable. We have now added this discussion to page 6 of the revised manuscript. It is copied here:

“We note that while the above two cases related to [emim][Tf₂N] solvation yield positive free energy values, it does not mean that [emim][Tf₂N] are not stable in these environments, since the free energy of a [emim][Tf₂N] in vacuum, which is the reference level, is 149.5 kJ/mol.”

4) The statement "In summary, we have demonstrated that DSE can effectively remove salt ions from water by using [emim][Tf₂N]" is inaccurate. Strictly speaking, DSE did not remove salt from water, but rather remove water from aqueous saline solution. Salt extraction and water extraction are in fact two fundamentally different processes in the field of desalination.

Reply: We thank the reviewer for pointing out this problem. We have modified the sentence to:

“In summary, we have demonstrated that DSE can effectively extract pure water from saline sources, even saturated saline sources, by using [emim][Tf₂N].”

5) Finally, it is suggested that the authors provide a concise overall discussion on DSE technology not only highlighting future opportunities but also the remaining challenges and limitations of the start-of-the-art development including the work presented in this manuscript.

Reply: We thank the reviewer for this suggestion, and we have updated the manuscript with a concise discussion of future research needs (at the end of the Discussion section). It is also copied below:

“The work is also expected to further stimulate research around the DSE technology, especially the exploration of even higher-performance DSs. To move DSE further closer to practical deployment, mass production of the identified ILs, process optimization, thermal system design with the potential integration of waste heat or renewable energy are topics worth additional research or engineering.”

Other General Edits:

In SI under general methods “N-Trifluoromethanesulfonylleucine” was changed to *N*-trifluoromethanesulfonylleucine.

Added funding agencies (NSF CHE-1956170 and NSF CBET-2031431) in acknowledgements.

References:

1. Jorgensen, W. L., Chandrasekhar, J., Madura, J. D., Impey, R. W. & Klein, M. L. Comparison of simple potential functions for simulating liquid water. *J. Chem. Phys.* 79, 926-935 (1983).
2. Bajpayee, A., Luo, T., Muto, A. & Chen, G. Very low temperature membrane-free desalination by directional solvent extraction. *Energy & Environmental Science* 4, 1672-1675 (2011).
3. T.Luo, A. B., G.Chen. Directional solvent for membrane-free water desalination—A molecular level study. *Journal of Applied Physics* 110, 054905 (2011).
4. Elimelech, M. & Phillip, W. A. The future of seawater desalination: energy, technology, and the environment. *Science* 333, 712-717 (2011).
5. Alhazmy, M. M. Feed water cooler to increase evaporation range in MSF plants. *Energy* 34, 7-13 (2009).

REVIEWER COMMENTS

Reviewer #1 (Remarks to the Author):

I praise the authors for carefully addressing reviewers comments and conducting additional simulations to provide in depth understanding and validation. The only issue remained unsolved in my opinion is the confusion regarding free energy of solvation as pointed in item #3 by reviewer 3 in previous round. Specifically, the authors report highly positive values for free energy of solvation of IL "molecule" in pure IL or in water (Table S5). It certainly does not make sense that IL solvation is highly unfavorable in itself. But i believe this confusion is due to the selected reference state:

- 1) The author report the free energy of solvation of a "molecule". In case of EMIM-TFSI, what is the "molecule"? a single ion or ionic pair?
- 2) The authors mention that the solvation energy of IL in vacuum is even higher than in bulk IL. What does it mean solvation energy in vacuum? What is the 149.5 kJ/mol mentioned in the response letter is relative to? It seems to me that the authors calculated the energy difference between ionic pair in vacuum and two isolated ions in vacuum.
- 3) It will make more sense to report the free energy of solvation between two states: ionic pair in vacuum and same pair in bulk IL. If I interpret the discussion in the response letter and data in Table S5 correctly then the free energy of solvation of EMIM-TFSI pair in bulk IL will be around -38 kJ/mol. I would strongly suggest the authors to choose a different reference state and minimize the confusion for the readers.

Reviewer #2 (Remarks to the Author):

The authors have satisfactorily addressed the criticisms I made to the previous version of the ms., so I recommend it is accepted in Nat. Comm in its present form.

Dear Reviewers,

We appreciate the time and effort you put to the reviewing processes regarding our manuscript, "Ionic Liquid Enables Highly Efficient Low Temperature Desalination by Directional Solvent Extraction", by Jiayi Guo, Zachary Tucker, Yu Wang, Brandon Ashfeld and Tengfei Luo. We have carefully studied the reviewers' comments and included our point-by-point responses below. We have also modified the manuscript accordingly, and the changes are highlighted.

Reviewer 1

I praise the authors for carefully addressing reviewers' comments and conducting additional simulations to provide in depth understanding and validation. The only issue remained unsolved in my opinion is the confusion regarding free energy of solvation as pointed in item #3 by reviewer 3 in previous round. Specifically, the authors report highly positive values for free energy of solvation of IL "molecule" in pure IL or in water (Table S5). It certainly does not make sense that IL solvation is highly unfavorable in itself. But I believe this confusion is due to the selected reference state:

1) The authors report the free energy of solvation of a "molecule". In case of EMIM-TFSI, what is the "molecule"? a single ion or ionic pair?

Response: Indeed, as the reviewer suspected, the "molecule" means the ionic pair. We have made this explicit in the main text (see our response to comment 3).

2) The authors mention that the solvation energy of IL in vacuum is even higher than in bulk IL. What does it mean solvation energy in vacuum? What is the 149.5 kJ/mol mentioned in the response letter is relative to? It seems to me that the authors calculated the energy difference between ionic pair in vacuum and two isolated ions in vacuum.

Response: We thank the reviewer for these questions. The solvation free energy calculated is essentially the Gibbs free energy needed to "appear" a solute into a solvent. For solvation energy in vacuum, it means that the solute molecule is "appearing" in vacuum. The free energy change of this process is due to the interactions between the two ions. So, this calculated free energy difference is indeed between ionic pair in vacuum and two isolated ions in vacuum. We have made necessary changes to the manuscript (see comment 3).

3) It will make more sense to report the free energy of solvation between two states: ionic pair in vacuum and same pair in bulk IL. If I interpret the discussion in the response letter and data in Table S5 correctly then the free energy of solvation of EMIM-TFSI pair in bulk IL will be around -38 kJ/mol. I would strongly suggest the authors to choose a different reference state and minimize the confusion for the readers.

Response: This is a great point, and that was actually why we calculated the free energy in vacuum. Since all calculations involving EMIM-TFSI as the solute used the same reference, which is "two isolated ions in vacuum" as mentioned in the above comment, the comparison was fair. As suggested by the reviewer, it might indeed be more straight forward if we use the ionic pair in vacuum as the reference. We have thus modified the manuscript to reflect this change. In the main text, we have changed the relevant sentence to:

“Additionally, the calculated solvation free energy of -38.6 kJ/mol for [emim][Tf₂N] in itself is significantly less than that of [emim][Tf₂N] in water (-23.3 kJ/mol), which suggests that it is thermodynamically unfavorable for the IL to dissolve in water. We note that the above two cases related to [emim][Tf₂N] solvation used the state of a [emim][Tf₂N] ionic pair in vacuum as the reference level.”

In the SI, Table S5 is modified to:

“**Table S5.** Free energy of solvation from simulation at 350K (77°C)

solute	solvent	free energy of solvation (kJ/mol)
TIP4P	TIP4P	-22.9
TIP4P	[emim][Tf ₂ N]	-26.5
[emim][Tf ₂ N]	TIP4P	-23.3*
[emim][Tf ₂ N]	[emim][Tf ₂ N]	-38.6*
NaCl	TIP4P	-699.5
NaCl	[emim][Tf ₂ N]	-677.7
TIP3P	TIP3P	-23.9
TIP3P	[emim][Tf ₂ N]	-24.0
NaCl	TIP3P	-709.6
[emim][Tf ₂ N]	TIP3P	-15.1*

* solvation free energy calculated involving [emim][Tf₂N] as a solute used ionic pair in vacuum (149.5 kJ/mol) as the reference state.”

REVIEWERS' COMMENTS<

Reviewer #1 (Remarks to the Author):

The authors addressed all my concerns and I believed that implemented modifications have clarified the previous confusion with the data. I believe the manuscript is publishable in current form.